# Readability of Commonly Used Quality of Life Outcome Measures for Youth Self-Report

**DOI:** 10.3390/ijerph19159555

**Published:** 2022-08-03

**Authors:** Karolin R. Krause, Jenna Jacob, Peter Szatmari, Daniel Hayes

**Affiliations:** 1Cundill Centre for Child and Youth Depression, Centre for Addiction and Mental Health (CAMH), 80 Workman Way, Toronto, ON M6J 1H4, Canada; karolin.krause@camh.ca (K.R.K.); peter.szatmari@utoronto.ca (P.S.); 2Research Department of Clinical, Educational and Health Psychology, University College London, Gower Street, London WC1E 6BT, UK; daniel.hayes@annafreud.org; 3Applied Research and Evaluation, Anna Freud National Centre for Children and Families, 4-8 Rodney Street, London N1 9JH, UK; 4Department of Psychiatry, The Hospital for Sick Children, Toronto, ON M5G 1X8, Canada; 5Department of Psychiatry, University of Toronto Faculty of Medicine, Toronto, ON M5S 1A8, Canada; 6Research Department of Behavioural Science and Health, Institute of Epidemiology & Health Care, University College London, Torrington Place, London WC1E 7HB, UK

**Keywords:** quality of life, children, youth, adolescents, outcome measures, readability

## Abstract

Self-report measures are central in capturing young people’s perspectives on mental health concerns and treatment outcomes. For children and adolescents to complete such measures meaningfully and independently, the reading difficulty must match their reading ability. Prior research suggests a frequent mismatch for mental health symptom measures. Similar analyses are lacking for measures of Quality of Life (QoL). We analysed the readability of 13 commonly used QoL self-report measures for children and adolescents aged 6 to 18 years by computing five readability formulas and a mean reading age across formulas. Across measures, the mean reading age for item sets was 10.7 years (SD = 1.2). For almost two-thirds of the questionnaires, the required reading age exceeded the minimum age of the target group by at least one year, with an average discrepancy of 3.0 years (SD = 1.2). Questionnaires with matching reading ages primarily targeted adolescents. Our study suggests a frequent mismatch between the reading difficulty of QoL self-report measures for pre-adolescent children and this group’s expected reading ability. Such discrepancies risk undermining the validity of measurement, especially where children also have learning or attention difficulties. Readability should be critically considered in measure development, as one aspect of the content validity of self-report measures for youth.

## 1. Introduction

With the majority of adults across 32 countries demonstrating a reading age of between 7 and 11 years old (33.3% at 7–9 years and 38.2% at 9–11 years old) [1], it is evident that reading age does not necessarily equate to chronological age. It is therefore important to ensure that the reading age of questionnaires used to measure health outcomes is appropriate for the population they are meant to be completed by. This is particularly important for mental health measurement with children and adolescents, whose reading levels may vary with age and with the presence of any learning difficulties.

Readability is a core aspect of a text’s broader comprehensibility, alongside other factors such as the organisation, layout, and structure; the accessibility of vocabulary; the presentation of numerical data (e.g., using phrases such as “two thirds” as opposed to percentages); and aspects related to the clarity and transparency with which arguments or intentions are articulated [2]. Questionnaires typically do not present complex narratives, arguments, or data; however, readability (alongside organisation, layout, and structure) is central to their comprehensibility.

Young people have raised challenges associated with the readability and broader comprehensibility of mental health and wellbeing measures, with issues including unfamiliar words; ambiguous wording; and complex, double-barrelled sentences [3]. Commonly used youth mental health measures have been found to often not be age appropriate with regards to their required reading level [4]. The age ranges of measures are often dictated by an assumed target and the age of the samples on which the initial psychometric testing was conducted. Initial scale development and validation rarely include readability assessments, despite this being a recommendation for best practice for the development of quality of life (QoL) measures (e.g., Ref. [5]). The COnsensus-based Standards for the selection of health Measurement Instruments (COSMIN) suggest that a measure’s comprehensibility should be assessed as one of three core criteria of content validity (alongside relevance and comprehensiveness) [6], but the COSMIN manual for assessing content validity does not make specific recommendations as to how readability—as a core aspect of comprehensibility—should be tested [6,7].

In child and adolescent mental health, self-report measures have a vital role in capturing initial concerns and progress in treatment outcomes from young people’s own perspectives. Compared with reports by parents, carers, or clinicians, young peoples’ self-reporting offers a unique and complementary perspective on mental health difficulties and on how these difficulties affect daily life [8,9,10]. Discrepant reports between these different informants not only reveal different perspectives and priorities [11] but can also convey information about the different contexts in which mental health difficulties are most manifested [12]. Finally, the inclusion of youth voices is a core principle of an ongoing transformative movement towards more person-centred and youth-friendly models of mental health care [13,14]. It is therefore of paramount importance to ensure that the voices of children and adolescents are centred in the measurement of mental health concerns and outcomes, enabling them to have appropriate understanding, involvement, and empowerment in their care [15,16]. This can only work effectively if children and adolescents are able to understand the questions asked of them and can read measures without input from others.

There is a strong ethical responsibility to ensure that measures are suitable and accessible to children and adolescents. If young people require assistance to understand measures, this may negatively affect their self-efficacy and engagement with the measurement process. There are further links to emotional wellbeing and literacy skills [17], such that reduced wellbeing has connotations for a young person’s ability to effectively learn, thus further highlighting the need to ensure measures are at a suitable reading age. Children and adolescents from minoritized ethnic groups are likely to be particularly impacted by measures with high required reading levels if English is a second language and/or if young people are less familiar with the language and culture of the majority of society’s psychological assessment procedures [18]. Young people often do not complete a full course of mental health treatment [19], and poor therapeutic alliance has been identified as one contributing factor [20]. Although underexplored, undue measurement burden may also play a role in this context. Finally, a poor fit between a measure’s readability and the respondents’ reading age can affect a measure’s validity and reliability. Items that are difficult to understand will likely have higher rates of non-response [21]. Where young people require help from others, they may be more likely to guess “adequate” responses to hard-to-understand questions to please the helper or to appear capable [22].

Previous studies have assessed the readability of instruments measuring the symptoms of mental health conditions, or screening for psychosocial difficulties more broadly speaking [4,23]. They suggest that such measures are often too difficult to read for their intended target group. Another important outcome domain relates to the impact that mental health symptoms are having on children and adolescents’ daily lives [24]. The International Consortium for Health Outcomes Measurement (ICHOM) recently recommended that functioning should be measured as a core outcome, alongside symptoms, for all young people seeking treatment for anxiety or depression [25]. The choice of self-report measures of functioning is, however, limited, with many commonly used instruments being parent- or clinician-rated [26]. Consequently, one of the measures recommended by ICHOM to assess the life impact of mental health difficulties is a QoL measure (i.e., the KIDSCREEN 10) [26]. QoL can be understood as a broader construct within which functioning is nested [25,27].

Given the growing interest in assessing the impact of mental health difficulties on young people’s daily life, and the vital role of QoL measures for capturing young people’s self-report in this domain, there is value in examining the readability of relevant instruments. The present study will explore the readability of commonly used youth self-report outcome measures of QoL, which have previously been identified through a scoping umbrella review [24]. We will use several established readability formulas that are designed to help predict the difficulty associated with reading any given text [28].

## 2. Materials and Methods

### 2.1. Selection of QoL Measures

We identified measures to include in this study as part of a scoping umbrella review that mapped measures of functioning, QoL, and wellbeing applied in mental health contexts with young people aged six to 24 years. The review methodology and primary findings have been reported elsewhere [24,25]. In short, a systematic search across six bibliographic databases (MEDLINE, Embase, APA PsycINFO, CINAHL, Web of Science and the COSMIN database of systematic reviews of measurement instruments) identified over 7400 unique records that were systematically screened for eligibility. We included 39 review articles and extracted measures assessing functioning, QoL, or wellbeing.

To be considered for inclusion in the present study, measures had to: (a) assess generic QoL (as opposed to disease-specific QoL) as a target construct according to the measure’s original development or validation study; (b) be developed specifically for children and/or adolescents between six and 18 years old (i.e., measures developed for an adult population and later validated with young people were not eligible as we considered that their readability might exceed the reading levels of children and/or adolescents by design; however, measures *adapted* for use with children and/or adolescents were eligible); (c) be available as a self-report questionnaire in English; and (d) be freely available for the authors to review. Given that the previous scoping review identified a large number of QoL measures [24], we chose to focus this study on the most highly cited measures that met the above inclusion criteria (i.e., ≥50 citations). Citations were determined for the manuscript reporting the initial development or validation study, using Google Scholar [24]. For each of the included QoL measures, we identified all age-specific versions, as well as short and long forms.

### 2.2. Readability Formulas

A variety of formulas are available to quantify the ease of reading comprehension based on text analysis. Different formulas use different criteria such as the number of syllables per word or per sentence, the occurrence of words with multiple syllables, or the number of letters per word (see Table 1). We chose to triangulate five formulas that complement each other by providing different approaches to calculating readability. We also present correlations between these five formulas and a mean readability score across all formulas. For ease of interpretation, we converted results from reading grade to reading age by adding five years onto the grade level yielded by the relevant formula, which corresponds to grade-specific ages in North America.

### 2.3. Coleman-Liau Index (CLI)

The CLI was developed as a readability formula that lends itself to machine-assisted scoring [29]. The CLI yields a US school grade reading level based on the average number of letters per 100 words and the average number of sentences per 100 words.

### 2.4. Flesch Kincaid (FK)

The FK [30] was adapted based on the Flesch Reading Ease [31] and represents one of the oldest and most widely used readability statistics. The FK is calculated based on the average number of syllables per word and sentence length and produces an estimated US grade level.

### 2.5. FORCAST

The FORCAST formula [32] was specifically developed for a text that is not prose, such as questionnaires, forms, lists, or websites [33]. Contrary to the above-mentioned formulas, FORCAST disregards the number or length of sentences, which can help avoid bias arising from short sentences or lack of punctuation in questionnaires. A reading level is calculated based on the number of monosyllabic words. FORCAST has been shown to correlate highly with other readability formulas [32] and has been validated against measures of reading comprehension [34].

### 2.6. Gunning Fog Index (FOG)

Like the FRE, the FOG also considers syllables and sentence length [35]. More specifically, the FOG considers the average number of words per sentence, and the incidence of difficult words, defined as words consisting of three or more syllables.

### 2.7. Dale-Chall Readability Formula (DC)

The DC assesses vocabulary rather than text difficulty [36]. Reading ease is established by considering the percentage of difficult vocabulary present in a text, defined as words that do not appear on a list of around 3000 easy-to-understand words that up to 80% of US fourth graders are expected to be familiar with. We created an amended Dale-Chall list by adding plural forms in addition to singular nouns (e.g., “things” in addition to “thing”; “kids” in addition to “kid”) as well as past tense, third-person present tense, and gerund forms to verbs (e.g., “writing”, “wrote”, “writes” in addition to “write”). We made this modification based on the assumption that these derivations would not be considerably less familiar than the stem term included in the original Dale-Chall list.

### 2.8. Readability Analysis

We conducted separate readability analyses for questionnaire instructions (including the introductory question/paragraph, example items, and any instructions provided in between items); questionnaire item sets assessing QoL; and any additional items assessing respondent demographic characteristics (presented in Appendix A). In line with earlier, similar studies, we standardised the questionnaire text by placing periods at the end of items, and by removing question numbers [4]. The decision to remove question numbers was made because the CLI, in particular, proved highly sensitive to the inclusion of question numbering, and scores were more consistent with other formulas when numbering was removed. We only included response options if they were provided as full sentences, instead of numerical or Likert-type response categories consisting only of a few words [37]. However, where some response sets within the same questionnaire consisted of full sentences while other response sets did not, we still included the latter to maintain a consistent approach within the same questionnaire. We also included response options if they formed a sentence in combination with the item.

All analyses were conducted using the KoRpus text analysis package in R [38]. For several formulas, KoRpus converts US grade levels into a reading age by adding five to a grade-level score. For formulas where KoRpus does not automatically provide this conversion, we added five to the reported grade level, to be able to consistently report results as reading age. For the DC formula, KoRpus computes a target age range (e.g., 10–12 years). To obtain a point estimate to consider in the computation of the mean reading age across formulas, we used the percentage of difficult words as indicated by KoRpus, to calculate a mean reading age manually using Power’s updated Dale-Chall formula [(3.2672 + (0.0596 × sentence length) + (0.1155 × % of difficult words)] [39].

## 3. Results

We included 13 self-report measures in this review, including the PROMIS questionnaires assessing Paediatric Global Health. While the latter assess general self-rated health rather than QoL, increasing interest in the system of PROMIS measures as a resource to help with harmonising the use of self-report measures across different conditions and outcome domains led us to include these alongside designated QoL measures [40].

Of the 13 measures included in this review, 10 had one questionnaire version available, three had two questionnaire versions available, one had three versions available, and one had four versions available (see Table 2). Consequently, we analysed 21 questionnaires across the 13 QoL measures. Of the 21 individual questionnaires, four were designed primarily for use with pre-adolescent children (i.e., with the target age ranging from seven or eight years through 11 or 12 years, depending on the measure), eight were designed primarily for use with adolescents (i.e., with the target age starting at 11 years), and nine were designed for use with children and adolescents (i.e., with the target age starting at six years and extending up to at least 15 years). Table 2 displays the questionnaires, and the required minimum reading age for the questionnaire instructions and item sets, according to the CLI, DC, FK, FOG, and FORCAST, as well as the mean reading age across all formulas. Appendix A shows the text characteristics (i.e., number of words, number of sentences, average syllables per word, average words per sentence, percentage of difficult vocabulary).

### 3.1. Correlations among Readability Formulas

For the questionnaire instructions, the CLI and FK showed moderate to strong, significant (*p* < 0.05) correlations with all other readability formulas, ranging from r = 0.46 to r = 0.92. Correlations between the FORCAST and the FOG, and the FORCAST and the DC were moderate (r = 0.49 to 0.65, *p* < 0.05). The correlation between the DC and the FOG was non-significant. Low or non-significant correlations may be due to the limitations associated with applying these formulas to short text passages, as reliability and consistency between formulas increase with text length [41]. On average, across all questionnaire instructions, the FK showed the lowest reading age (9.3 years, SD = 2.0) while the FORCAST showed the highest reading age (13.8 years, SD = 1.0).

For the item sets, we observed strong correlations between the FK on the one hand, and the FOG, FORCAST, and CLI on the other hand (r = 0.71 to 0.82, *p* < 0.001), as well as between the CLI and the FOG (r = 0.78, *p* < 0.001). Correlations were moderate for the CLI and the FORCAST (r = 0.52), the DC and the FK (r = 0.48), and the DC and the FORCAST (r = 0.56), with *p* < 0.05. Correlations were insignificant for the CLI and DC, the DC and FOG, and the FOG and FORCAST. On average across all questionnaire item sets, the CLI showed the lowest reading age (8.6 years, SD = 1.6) while the FORCAST showed the highest reading age (14.0 years, SD = 1.0).

### 3.2. Instructions

Of 21 questionnaires, 20 had instructions available. On average, instructions included 166.4 words (SD = 264.2, range = 17–1253 words) across an average of 13.9 sentences (SD = 17.2; range = 2–82). The mean reading age for the instructions across all readability formulas was 11.3 years (SD = 1.6, range = 9.9–15.9 years). 

For the four questionnaires that were designed primarily for pre-adolescent children, the mean reading age for the instructions across all formulas was 10.4 years (SD = 0.5, range = 10.0–11.2). For the eight questionnaires that were designed primarily for adolescents and had instructions available, the mean reading age for the instructions was 12.0 years (SD = 2.0, range = 10.0–15.9 years). For the eight measures that targeted children and adolescents and had instructions available, the mean reading age for the instructions was 11.0 years (SD = 1.2, range = 9.9–12.8 years).

For 75.0% of the questionnaires, the required reading age exceeded the minimum indicated target age by at least one year. The average discrepancy between the indicated lower target age and the required reading level for these questionnaires was 2.8 years (SD = 1.0, range = −4.8 to −1.6 years). The questionnaires that showed no or minimal discrepancies between target age and required reading level were the Kiddo-Kindlr [42,43], the PedsQL long and short forms for teens [44], the PQ-LES-Q [45], the QOLP-AV [46], and the 16D [47], all of which are questionnaires targeting primarily adolescents. 

### 3.3. Items

The 21 individual questionnaires associated with the 13 included measures had an average word count of 393.3 words (SD = 349.7; range = 65–1024 words) across an average of 49.2 sentences (SD = 45.5; range = 7–172 sentences). The mean reading age for the item sets across all readability formulas was 10.7 years (SD = 1.2, range = 9.1–12.4 years). 

For the four questionnaires that were designed primarily for pre-adolescent children, the mean reading age across formulas was 9.6 years (SD = 0.8, range = 9.1–10.9 years). For the eight questionnaires that were designed primarily for adolescents, the mean reading age was 10.9 years (SD = 1.4, range = 9.1–12.1 years). For the nine questionnaires that were targeting children and adolescents, the mean reading age was 11.1 years (SD = 1.0, range = 10.1–12.4 years).

For 13 out of 21 questionnaires (61.9%), the required reading age for the item set exceeded the minimum indicated target age for the given questionnaire by at least one year. The average discrepancy between the indicated lower target age and the required reading level for these questionnaires was 3.0 years (SD = 1.2, range = −4.4 to −1.2 years). The questionnaires that showed no discrepancy between target age and required reading level for the item set were the AQOL-6D [48], the Kiddo-Kindlr [42,43], the PedsQL long and short forms for teens [44], the QOLP-AV [46], the YQOL long and short form [49], and the 16D [47], all of which were designed primarily for adolescents.

### 3.4. Associations between Questionnaire Length and Readability

There was no significant correlation between the word count of the questionnaire instructions and the required minimum reading age. There was, however, a moderate correlation between the length of item sets and the required reading age (r = 0.44, *p* = 0.04). This suggests that longer item sets did tend to be more difficult to read than shorter item sets. One measure (the QOLP-AV) was excluded from the analysis pertaining to instruction length, due to instructions of outlying length (1253 words).

**Table 2 ijerph-19-09555-t002:** Reading Age for Commonly Used QoL Measures.

				Instructions						Items					
Measure	Ages	Length	Citations	CLI	DC	FK	FOG	FCST	Mean	CLI	DC	FK	FOG	FCST	Mean
AQOL-6D Adolescent Instrument [48]	12–18 years	20 items	89	16.8	12.2	15.8	18.9	15.6	15.9	11.8	10.8	10.4	12.7	14.9	12.1
CHU-9D [50]	7–17 years	9 items	230	8.6	9.1	8.3	8.9	14.6	9.9	8.8	10.6	8.4	8.4	15.2	10.3
EQ-5D-Y [51]	8–15 years	6 items	560	9.5	10.5	7.5	10.4	12.1	10.0	11.4	10.9	10.9	12.5	16.5	12.4
KIDSCREEN—*KIDSCREEN-52* [52]	8–18 years	52 items	827	10.3	9.4	8.1	11.2	12.9	10.4	10.4	9.8	7.7	9.5	13.3	10.1
KIDSCREEN—*KIDSCREEN-27* [53]	8–18 years	27 items	612	10.3	9.4	8.1	11.2	12.9	10.4	10.2	10.1	8.1	10.1	13.2	10.3
KIDSCREEN—*KIDSCREEN-10* [54]	8–18 years	10 items	489	10.3	9.4	8.1	11.2	12.9	10.4	12.3	9.2	6.4	11.2	12.9	10.4
KINDL^R^—*Kid-KINDL^R^* [42,43]	7–13 years	24 items	1218	10.7	9.5	7.8	9.9	13.4	10.3	7.2	10.1	6.6	8.7	13.1	9.1
KINDL^R^—*Kiddo-KINDL^R^* [42,43]	14–17 years	24 items	1218	10.7	9.5	7.8	9.8	13.3	10.2	7.0	10.1	6.8	9.0	13.0	9.2
PedsQL 4.0 *Generic Core Scales—Child* [44]	8–12 years	23 items	3269	7.6	11.4	8.5	10.0	13.3	10.2	7.0	9.6	7.0	9.1	13.1	9.2
PedsQL 4.0 *SF15 Generic Core Scales—Child* [44]	8–12 years	15 items	3269	7.6	10.6	8.5	10.0	13.3	10.0	7.2	9.9	7.1	9.0	13.3	9.3
PedsQL 4.0 *Generic Core Scales—Teen* [44]	13–18 years	23 items	3269	7.6	10.6	8.5	10.0	13.3	10.0	7.0	9.9	7.0	9.1	13.1	9.2
PedsQL 4.0 *SF15 Generic Core Scales—Teen* [44]	13–18 years	15 items	3269	7.6	10.6	8.5	10.0	13.3	10.0	7.3	10.1	7.1	9.0	13.3	9.4
PQ-LES-Q [45]	6–17 years	15 items	139	-	-	-	-	-	-	10.8	9.6	7.6	9.3	14.9	10.4
PROMIS Global Health 7 [55]	8–17 years	7 items	72	13.7	13.5	10.1	10.8	15.3	12.7	11.9	10.1	9.9	14.6	13.4	12.0
PROMIS Global Health 7 + 2 [55]	8–17 years	9 items	72	14.1	13.5	10.1	10.8	15.3	12.8	10.8	10.2	9.6	13.9	13.4	11.6
QOLP-AV [46]	14–20 years	72 items	280	12.4	11.9	12.6	15.0	14.2	13.2	10.6	11.4	9.7	12.2	15.2	11.8
TACQOL [56]	8–15 years	63 items	98	11.6	10.2	9.2	11.4	13.5	11.2	13.1	11.2	10.2	12.1	15.2	12.4
YQOL—*YQOL-R* [49]	11–18 years	57 items	282	14.3	10.6	11.0	12.5	14.8	12.6	11.9	10.1	10.5	12.1	14.5	11.8
YQOL—*YQOL-SF* [49]	11–18 years	16 items	282	14.1	10.6	11.1	13.0	14.7	12.7	11.6	10.0	10.6	12.3	14.5	11.8
16D [47]	12–15 years	16 items	201	12.7	9.8	9.0	12.1	13.5	11.4	11.7	10.1	10.3	12.5	14.7	11.9
17D [57]	8–11 years	17 items	179	12.7	10.3	8.3	10.8	13.9	11.2	9.6	11.9	8.3	10.6	14.0	10.9
Overall mean				11.2	10.6	9.3	11.4	13.8	11.3	10.0	10.3	8.6	10.8	14.0	10.7
(SD)				(2.6)	(1.3)	(2.0)	(2.2)	(1.0)	(1.6)	(2.1)	(0.6)	(1.6)	(1.9)	(1.0)	(1.2)

Note: AQoL-6D: Assessment of Quality of Life; CHU-9D: Child Health Utility Index 9D; EQ-5D-Y: EuroQol Five Dimensions Health for Youth; HUI2: Health Utilities Index Mark 2; HUI3: Health Utilities Index Mark 3; KINDLR: KINDer Lebensqualitätsfragebogen; PedsQL 4.0: Pediatric Quality of Life Inventory Generic Core Scales; PQ-LES-Q: The Pediatric Quality of Life Enjoyment and Satisfaction Questionnaire; PROMIS: the Patient Reported Outcomes Measurement Information System; QOLP-AV: Quality of Life Profile—Adolescent Version; TACQOL: TNO AZL Children’s Quality of Life; YQOL: Youth Quality of Life Instrument; YQOL-R: Youth Quality of Life Instrument—Research Version; YQOL-SF: Youth Quality of Life Instrument—Short Form; SF15: Short Form 15.

## 4. Discussion

This was the first study to compare the readability of 13 commonly used outcome measures of QoL designed for self-completion by children and/or adolescents. We considered 21 individual questionnaires across 13 measures (e.g., including short and age-specific versions). The mean reading age was 11.2 years for questionnaire instructions, and 10.7 years for questionnaire items. This reading age exceeded the questionnaire’s minimum target age by at least one year for 15 out of 20 available instructions, and 13 out of 21 item sets. All questionnaires for which the reading age did not exceed the minimum target age were designed for adolescents. There was no significant association between instruction length and reading difficulty, but we did observe a moderate correlation between the length of item sets and reading difficulty. This suggests that shorter questionnaires may have a tendency of being easier to read.

Our findings add to existing research that highlights discrepancies between the age group targeted by measures used in youth mental health, and the required reading age of these measures. In a readability analysis considering 35 mental health symptom measures for child and adolescent self-report, Jensen and colleagues [4] computed an average required reading grade of 6.35, equivalent to a reading age of 11.4 years. The authors did not systematically compare the reading level of individual measures to the minimum age of the target population but concluded that “reading levels required to accurately read and comprehend several common child-, adolescent-, and parent-report measures are often higher than the reading ability of the rater” (p. 350, [4]). Patalay and colleagues [23] examined the readability of the Strengths and Difficulties Questionnaire (SDQ), a widely used self-reported screening tool for psychosocial difficulties in young people aged 11–17 years [58] both as a whole and via each of the measure’s subscales. Although two of the SDQ’s five subscales and the overall composite measure had a minimum reading age of 11–12 years, the instructions and three other subscales required a higher reading age of 13–14 years.

It has been suggested that health questionnaires designed for adults should have a fifth-to-sixth grade reading level, equivalent to a reading age of 10–11 years of age [59]. In the present study, the mean reading age of questionnaires designed primarily for pre-adolescent children was 10.4 years for the instructions and 9.6 years for the items. The mean reading age of questionnaires designed for adolescents was 12.0 years for the instructions and 10.9 years for the items. QoL measures designed for children and adolescents hence require roughly the same reading level that is recommended for adult health questionnaires and may not make effective concessions to the literacy levels of their younger target groups. Our finding whereby all questionnaires that matched their target age group in terms of readability were designed for adolescents, suggests that QoL questionnaires designed for pre-adolescent children may typically be too difficult to read for the younger end of this age bracket. This is of heightened concern when using QoL measures in mental health contexts where young people may be over-proportionately affected by learning difficulties and special educational needs [23,60].

Some of the measures we considered were adapted for use with children or adolescents from an original adult measure. These include the EQ-5D-Y [51] and the AQOL-6D [48]. It could be expected that such measures are even more prone to poor readability than measures developed specifically for children and young people. We found this to be the case for the EQ-5D-Y but not for the AQOL-6D.

Our readability analysis suggests that several QoL measures designed for children and/or adolescents might benefit from text revisions to match the reading ability of their youngest target group. Future efforts to revise, adapt, or translate these measures should consider ways of reducing reading difficulty. This could include keeping questions and sentences as short as possible, favouring short over long words with similar meaning, as well as identifying unfamiliar words (i.e., that are not on the Dale-Chall list) and replacing them with more familiar terms as available [3]. Similar considerations apply when designing new measures. Readability formulas offer a relatively quick way of gathering feedback on whether such efforts are successful in reducing formal reading difficulty. In addition, new questionnaires and questionnaire adaptations should be carefully piloted in all relevant age groups, and this should involve cognitive debriefing with specific consideration of readability [3]. However, measure adaption in mental health research is challenging, as any changes made to measures that are already in use can affect data comparability [61]. Hence, where the post-hoc simplification of questionnaire text is not possible, limitations around readability should be communicated when reporting data collected with relevant measures.

Our findings have important implications for the use of QoL self-report measures in clinical practice and in research studies. In clinical practice, young people may be asked to complete QoL self-report measures as part of measurement-based care, where individual outcome data are used to inform clinical decisions and treatment adjustments [62]. Children or adolescents may be asked to complete a measure during their care session, in which case clinicians may be able to help with reading comprehension. Alternatively, young people may be asked to complete a measure prior to their clinical appointment, at home or in the waiting room, where they may not be able to access help in reading a questionnaire or be subject to influences from adults if they request support in answering. As interest in measurement-based care is growing [63], and the session-by-session completion of outcome measures is increasingly encouraged (e.g., Ref. [25]), it is critical to carefully consider the respondent burden arising from the poor readability of questionnaires to prevent frustration, non-response, response bias, over or underreporting of symptoms, and treatment drop-out. In combination with varying levels of literacy and acculturation among different social, cultural, and ethnic groups of young people accessing mental health services, the possible consequences of poor readability may increase health disparities. Additional research is needed to understand the impact of poor readability (and the comprehensibility and content validity of outcome measures more broadly speaking) on young people’s treatment experiences, satisfaction, and adherence.

In research, a mismatch between a measure’s readability and the reading ability of study participants can undermine a measure’s comprehensibility and thus content validity [63]. It may also affect other measurement properties such as test-retest reliability if young people are guessing the meaning of items and if guesses evolve with repeated administration; or construct validity if scores on difficult-to-read measures are compared with measures that are easier to read or completed by a different informant. A dedicated effort may be needed to assess the reading ability of service users or research participants and compare this to the reading level of any measure used. Where there is a mismatch, options include the choice of a different measure, and the more careful interpretation of data collected [59]. Alternatively, where measures are completed online, support for difficult-to-read words could be introduced by providing alternatives when the curser is scrolled over such words.

### Limitations

Several important limitations must be noted. It has been suggested that readability formulas yield the most accurate results when applied to prose passages and that they may be less accurate when applied to the often short sentences constituting questionnaire items [41,64]. Readability formulas also tend to require text passages of at least 100 words to estimate readability reliably [65]. In our study, most item sets exceeded 100 words, although four questionnaires included less than 100 words (ranging from 65 words for the PQ-LES-Q to 98 words for PROMIS-7-Plus). Of the 20 available instructions, however, 11 included less than 100 words (ranging from 17 words for the two PROMIS questionnaires to 95 words for the CHU-9D). 

Readability statistics do not capture the overall comprehensibility of a document. They do not, for example, consider its visual organisation (e.g., font type and size, illustrations, white space). Many formulas were developed several decades ago and do not reflect recent changes in formatting such as the use of headings and subheadings, bullet points, or the use of numbering [41]. Readability formulas also do not assess the complexity of the constructs and ideas conveyed by a text or required points of references that a reader would need to be familiar with in order to correctly interpret a text’s meaning [28]. Other factors that are not accounted for are ambiguities inherent in the choice of specific terms [65], and clarifications that may arise from the combination of different terms or items [21]. Other tools are available for broader assessments of a text’s comprehensibility. For example, the Suitability Assessment of Materials (SAM) checklist has been specifically designed to assess the content, literacy requirements, graphics, layout, typography, and cultural appropriateness of health education documents [66]. However, some of the SAM’s 22 items may not apply to the assessment of questionnaires rather than prose text (e.g., items related to content) and may require adaptation.

There are several sources of inconsistency when applying readability formulas to text passages. The same readability formulas have been shown to yield differential scores when computed via different software programs, due to subtle differences in analysis parameters [41,67]. These parameters include the definition and counting of syllables, words, or sentences; and the handling of punctuation marks and other text elements such as hyphens, digits, or abbreviations. Many original formulas do not specify how exactly these elements should be managed [41]. Different readability formulas also tend to compute different readability scores for the same text passages, as we have observed in our study [28,41,65]. Zhou and colleagues [41] found that five commonly used readability formulas showed an average discrepancy range of 2.3 grade levels. Consistency improved with the length of the tested passages, and the authors recommended using text samples between 500 and 900 words, or preferably, full documents to increase consistency. Jensen et al. [4] suggest that readability analyses have a restricted range of sensitivity and may not be sensitive to reading levels below the 4th grade (i.e., 9 years of age) or above the 12th grade (i.e., 17 years of age). They did not, however, elaborate or reference the evidence base for this suggestion. 

A further limitation of the present study is that we performed readability analyses in only one language, based on formulas developed primarily for use in US contexts [21]. There may be nuanced differences in readability across different English-speaking contexts. Several of the included measures are available in a number of other languages (e.g., the KIDSCREEN, the EQ-5D-Y), and assessing their readability in those languages through appropriate formulas would also be of interest.

Lastly, we examined the readability of ‘overall’ questionnaires, rather than of individual subscales. Some of the included QoL measures, such as the KIDSCREEN 52, consist of subscales pertaining to different constructs and previous research has found differences in subscale readability for the widely used SDQ [23]. Given that subscales are frequently used as stand-alone measures [3], future research should investigate differences in subscale readability, particularly, as it may be that minor modifications to particular subscales may be sufficient in lowering the overall reading age.

Despite these limitations, readability formulas provide an efficient means to quantify the reading difficulty of a given text [21]. Their use has been recommended to ensure that survey questions are appropriate for the reading ability of young people [68]. Content validity has been described as the most important measurement property of a patient-reported outcome measure (PROMS; [69]). The Consensus-based Standards for the selection of health Measurement Instruments (COSMIN) manual for assessing the content validity of PROMS suggests that a measure’s reading level should be considered when appraising content validity, alongside the presence of ambiguous items, double-barrelled questions, jargon, and valued-laden words [70]. Readability formulas thus represent one important tool amongst others (e.g., cognitive interviewing) to ensure the comprehensibility of QoL self-report measures.

## 5. Conclusions

Our study suggests that many commonly used self-report measures of QoL designed for children and adolescents are too difficult to read for young people at the younger end of the target age range for which they were designed. This is of particular concern for using such QoL measures in mental health settings, where young people may face additional barriers due to learning difficulties, as well as capacity issues due to both developmental stage and mental health diagnosis. It also raises equity concerns in relation to young people whose first language is not English. To ensure that young people can complete self-report measures meaningfully and autonomously, it is critical that the reading difficulty of these measures is proportionate. Therefore, readability should be a routine consideration when developing new QoL self-report questionnaires, and when appraising the content validity of existing measures. Although readability formulas such as those used in this study have limitations, they represent readily available tools that can be used as part of a multi-pronged approach to assessing comprehensibility as a core component of a questionnaire’s content validity.

## Figures and Tables

**Table 1 ijerph-19-09555-t001:** Information Used by Each Readability Formula.

	CLI	Dale-Chall	FK	FOG	FORCAST
Average number of letters per 100 words	X				
Average number of sentences per 100 words	X				
Average words per sentence		X	X	X	
Average syllables per word			X		
Incidence of words with 1 syllable					X
Incidence of words with >3 syllables				X	
Percentage of difficult vocabulary		X			

## Data Availability

Data sharing not applicable. The raw data analysed in this study (i.e., the analysed questionnaires) are licensed to the relevant measure developers or distributors and were made available to the authors under specific licensing terms. It is not the author’s right to publish these data. The analytical code used for the analyses presented in this manuscript is available in the Appendix A.

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
