# Peer review of "Readability of Commonly Used Quality of Life Outcome Measures for Youth Self-Report"

_ijerph, 2022, doi:10.3390/ijerph19159555_

Round 1
Reviewer 1 Report
This is an interesting paper on an important topic, i.e. the readability of quality of life measures designed specifically for youth/adolescents. The authors utilize 5 different readability measures to analyze 13 commonly used QoL measures. The results show that for 2/3 of these measures, the average reading age exceeds the minimum age specified.
I think this is an innovative methodology and an important topic; yet I do have some remarks for the authors.
- I think the authors should emphasize whether - and in particular, how - the readability of a measure/assessment instrument may impact on the comprehension of the text. At the moment, I feel the authors equate these concepts, whereas they are inherently different constructs (see page 3, where the authors state "[...] the ease of reading comprehension based on text analysis."). For example, the authors could have applied the Comprehensibility Assessment of Materials (SAM + CAM) scale. At a minimum, the authors should discuss how readability and comprehensibility differ, and why they chose to focus on only readability.
- The different assessment methods for readability result in very different age estimates - for some measures ranging between 6.6 and 13.1 years (the KID-Kindl). It would have been nice if the authors could have provided a test of the overall mean scores of these 5 formulas, to emphasize the fact that - for example - the FORCAST results in significantly higher estimates, and to discuss the utility of this formula in light of these differences.
- The authors mention that only Qol Measures specfically designed for youth were included in their study - why is this the case? I can imagine that existing measures that are adapted to be suitable for an adolescent target group may suffer even more from poor readability. The authors should discuss this issue somewhere.
- The authors do not really discuss the implications of their findings, nor do they offer solutions to improve the QoL measures they have analyzed. I would have expected a more thorough discussion of their findings, rather than a repetition of the arguments they also presented in the introduction section (on content validity etc.).
Reviewer 2 Report
The authors appropriately call attention to the relevance of readability of instruments used to measure the general quality of life (QoL) of children and adolescents, within the broader context of mental health measures. As the authors calim, such readability has a huge impact on the validity of these instruments as adequate measures for such populations.
The method used is adequate to respond to the research question and able to assess if the QoL measures assessed are indedd appropriate, in terms of reabability and understandability, to the age brackets of the youths they are measuring.
In fact, the results that are presented point to the fact that QoL questionnaires designed for pre-adolescent children may prove too difficult to read for the younger end of the targeted age bracket, considering both the items and the instructions. As the authors point out, such inappropriateness may have seroius consequences both for research and clinical practice.
Despite the reported limitations of the present study (e.g. readability statistics not capturing the overall accessibility of the questionnaires; results inconsistencies among studies/ analyses when applying readability formulas; performing the readability analyses in only one language; performing the readability analyses in overall questionnaires and not looking specifically at subscales), the fact of calling attention to the readability inappropriateness of QoL measures is in itself a relevant warning tp the research and clinical community, hopefully stimulating further studies aimed at correcting existing measures or producing other measures that are more in tune to the developmental parameters of the populations they target.
In sum, the present study has the merit of calling attention and quantitatively consubstantiate existing validity problems of some QoL measures (based readability issues), especially when applied to certain age brackets and, hence, stimulate future necessary corrections/ adaptations/ transformations of existing measures.
Assessing readability of a scale is an important factor that affects its validity. This issue is especially relevant with youth populations. When the average readability estimates are substantially higher than the age group (or part of it) to which a certain measure is designed to focus on, this may bring negative consequences both in terms of research and clinical assessment. This study has the merit of assessing QoL scales’ readability and pointing to the fact that some scales demand reading skills (both in terms of item content and instructions) that are higher than the mean reading skills of at least the younger portion of a certain age bracket to which the measure is being applied to. Calling attention to this fact and to the need to make these measures more adapted to the reading skills of the population they target is of paramount importance to enhance their validity and to avoid the negative consequences of incorrect assessment. This may be especially relevant when we take into consideration the fact that often times a sizable group of young people may have lower reading ages relative to their developmental age. This latter factor may be of particular relevance in clinical settings: in such contexts many children with some cognitive difficulties may also evidence learning difficulties and special educational needs that have an effect on their reading skills.
The introduction section comprehensively contextualizes the paper within the existing relevant literature and objectives are precisely stated.
In terms of methodology the paper gives a clear view of the procedures and statistical analyses undertaken, which are appropriate to answer the research questions.
The discussion underlines the relevance of the study itself and study findings. Especially relevant is the limitations section of the paper. Regarding this section, one can see that one of the most relevant limitations is the fact that the authors did not analyze the subscales of the measures that were object of the study. In fact, in the proposed article, the QoL scales’ readability was assessed for the whole scales. It would add value to the article if subscales were also assessed. As said, the authors indeed mention in the limitations of the study that they did not assess subscales. Yet, such an assessment would actually add more value to article, making it more informative and giving a more comprehensive assessment of the scales readability, eventually pointing to the most problematic subscales.
I would therefore urge authors to make such an assessment and report it in the article. This would mean that the article should be revised accordingly (major revision).
With respect to the English language and style I would just recommend a final style and minor spell check.
Reviewer 3 Report
The authors make an important contribution to the field of mental health from a technical perspective. Their analysis proposal (based on widely used formulas), results in an original proposal to the health field. They also acknowledge the limitations of the work and point out possible courses of action to be considered in future research.
The authors highlight the importance of self-reports in assessing the mental health of children and young people, emphasizing their readability. For this reason, they employ conventional readability formulas, which have been widely used in text assessment.
The research shows consistency between the problem statement and the methodology employed, which is very descriptive about the units of analysis and how they were approached. The analysis of the results is adequate to the type of information obtained, and discusses the outcomes in each type of self-report, showing that some of them have a level of readability above the age for which they are intended; on the other hand, other instruments coincide with the age for which they are intended.
However, the formulas present limitations of comparison, evaluation and interpretation, to assess the instructions of the self-reports; and as the authors themselves recognize there are several sources of inconsistency when applying readability formulas to inconvenient text passages. A broad review of these limitations and the scope of the results obtained are addressed in the discussion.
